# Exploitation of Plant Growth Promoting Bacteria for Sustainable Agriculture: Hierarchical Approach to Link Laboratory and Field Experiments

**DOI:** 10.3390/microorganisms10050865

**Published:** 2022-04-21

**Authors:** Federica Massa, Roberto Defez, Carmen Bianco

**Affiliations:** Institute of Biosciences and BioResources, Via P. Castellino 111, 80131 Naples, Italy; federica.massa@ibbr.cnr.it (F.M.); roberto.defez@ibbr.cnr.it (R.D.)

**Keywords:** climate change, plant and microbial communities biodiversity, PGPB, microbial competitiveness and persistence, field application

## Abstract

To feed a world population, which will reach 9.7 billion in 2050, agricultural production will have to increase by 35–56%. Therefore, more food is urgently needed. Yield improvements for any given crop would require adequate fertilizer, water, and plant protection from pests and disease, but their further abuse will be economically disadvantageous and will have a negative impact on the environment. Using even more agricultural inputs is simply not possible, and the availability of arable land will be increasingly reduced due to climate changes. To improve agricultural production without further consumption of natural resources, farmers have a powerful ally: the beneficial microorganisms inhabiting the rhizosphere. However, to fully exploit the benefits of these microorganisms and therefore to widely market microbial-based products, there are still gaps that need to be filled, and here we will describe some critical issues that should be better addressed.

## 1. Introduction

The beneficial microorganisms interacting with plants support their growth through various mechanisms: (i) improvement of nutrient availability; (ii) enhancement of root development; (iii) reduction of toxic compounds levels in the soil; (iv) improvement of resistance to both biotic and abiotic stresses [1]. Plants can grow around the Yellowstone’s hot springs, in deserts, under flooding, and in salty and contaminated soils thanks to their genetic plasticity but also to the contribution of microbial communities helping them overcome these challenges [2,3,4]. In addition, certain plant diseases seemed to be rarer in some plants, leading to hypothesize that the disease suppression could also be due to living microorganisms [5,6]. Studies carried out to date suggest that plants completely free of microorganisms represent an exception. Furthermore, in recent years, new beneficial microorganisms potentially applicable for agricultural production have been identified [7]. The best known and most ancient beneficial microorganisms are the Rhizobia. These microorganisms, described for the first time by Martin Beijerinck in 1888 [8], inhabit the roots of leguminous plants and provide them with fixed nitrogen. In the last few years, the knowledge of microorganisms’ diversity and molecular mechanisms that regulate plant–microorganism interactions has improved considerably [9]. The number of recognized microbes interacting with plants, including the beneficial ones, is large and continuously growing, and these microbes can be classified in different ways [10]. Beneficial microorganisms can act as free-living bacteria, symbiotic (that establish specific symbiotic relationships with plants, i.e., Rhizobia and *Frankia*), and endophytes (that colonize the interior plant tissues). Within them, a group of soil bacteria called plant growth-promoting bacteria (PGPB) stimulates the growth and health of the plant through different mechanisms [1]. These mechanisms include: (i) production of plant hormones; (ii) nitrogen fixation; (iii) solubilization of inorganic phosphate and mineralization of organic phosphate; (iv) antagonism against pathogens by production of antibiotics, enzymes, and fungicidal compounds; and (v) competition with detrimental microorganisms causing disease to crops. The effects of human-caused climate changes are becoming more and more evident as we see more record-breaking heat waves, intense droughts, shifts in rainfall patterns and an increase in average temperatures. These environmental changes touch every step of crop production [11,12]. Studies concerning the effects of climate changes on the living organisms inhabiting our planet have highlighted that in the future the strong acceleration of these changes will impact more and more on the growth and productivity of plants [13,14]. Furthermore, to cope with the increasing world population, an improved food production is required. However, this objective cannot be achieved by intensifying the exploitation of natural resources, which will be increasingly limited [15]. The application of efficient and low-cost PGPB, which need minimum external energy and chemicals, could be a valid strategy to improve the response of plants to stress and to make the best use of available natural resources [16,17]. Although considerable efforts have been made in recent years to clarify the mechanisms of action of these microorganisms, several aspects still remain to be clarified. Several factors can limit the success of PGPB and some of them are the following: (i) less than 2% of microbes interacting with plants are cultivable in the laboratory; (ii) PGPB products are often crop-specific; (iii) the effectiveness of PGPB is generally assessed using single microorganisms instead of an appropriate consortium and without clarifying the roles and relationships between the various microorganisms; (iv) PGPB performances are not evaluated under different climatic and agronomic conditions; (v) the stability and persistence of PGPB in the host plants are often time-dependent. The latter has many similarities with human gut microbial communities, which are closely interconnected with human health. Indeed, several strategic therapies to restore and/or maintain the eubiotic state of the microbial intestinal ecosystem are under investigation. In this perspective paper, we discuss and highlight some of the gaps preventing full exploitation of PGPB abilities for sustainable crop production.

## 2. Linking Plant Diversity with Microbial Diversity

The microbial community structure is modulated by various factors, whose influence varies according to the ecosystem in which the host plant grows [18]. The main drivers of microbial community diversity in agricultural ecosystems are: farming practices, environmental conditions, soil type, and plant species; whereas in natural ecosystems, these communities are mainly modulated by biotic interactions, plant species, and plant diversity [19]. Microbial communities and their host plants have closely linked identities: plant’s genotypic differences strongly influence the structure of plant microbial communities. To better explain this relationship and to analyse the effect of reduced plant genetic diversity on soil microbial abundance, the identification of plant candidate genes regulating the interaction with beneficial microbial communities could be an efficient strategy [20]. In species-rich plant communities, the aboveground litterfall and the belowground fine root mortality lead to the availability of greater amounts of carbon and nutrient resources for soil microorganisms. Microbial diversity and biomass are also stimulated by diverse root exudates, whose production is influenced by plant genotypes and development stages (seedling, vegetative, bolting, and flowering) [21]. The exudates released by roots might have a double effect: they stimulate the interaction with some particular microorganisms (i.e., beneficial microorganisms) and act as antimicrobial agents for other ones (i.e., pathogenic microorganisms). Therefore, the diversity and quantity of exudates released can influence the type of interaction that plants establish with the microorganisms with which they come into contact. On the other hand, interaction with PGPB can induce changes in the type of exudates, which in turn modulates the response of the host plant to various external stimuli [22]. In addition, different plant tissues can host distinct microbial communities [23]. For instance, since shoot tissues are exposed to challenging environmental factors as compared to roots, a high degree of adaptation is required for microbial colonization, and only a few microbial groups adapted to environmental variations will be successful. Moreover, it was observed that microorganisms, such as PGPB, containing key genes responsible for the health status of the host plant, are common in different plant species [24]. However, there is a knowledge gap concerning the taxonomy of PGPB, and this is probably due to the following reasons: (i) these microorganisms belongs to many different phyla, and most of their phylogenetic analyses have been carried out within their specific genus and not within a group; (ii) the microbial classification is initially carried out only evaluating the morphological, biochemical, and functional characteristics through culture-based methods, without considering the genetic features, thus leading to incorrect classifications of many taxa [24]. Therefore, a careful classification of microbial community associated with a given plant genotype is essential to recognize the efficiency of PGPB as bio-inoculants under various environmental conditions.

## 3. Competence and Persistence of Microorganisms in the Field: Effect of Inoculant Strains on the Resident Microflora

Various critical factors can influence the efficacy of PGPB-based bioinoculants, including soil health, colonization efficiency, and persistence in the soil [25]. One of the main factors influencing soil health is the carbon transformation [26]. The decomposition of plant residues and other organic matter, the cultivation intensity, and tillage lead to carbon transformation, which in turn directly affects the composition of microbial communities [27]. The efficiency of PGPB strongly depends on their ability to compete with autochthonous microorganisms present in the soil, and to adhere and colonize the external and/or internal part of plant tissues [28]. In these processes, the formation of biofilms by PGPB can play an important role. To assess the ability of PGPB to colonize roots and persist in the soils, several techniques can be used, including microbial enumerations by culture-based methods, DNA-based methods, and microscopy-based techniques. To ensure good results in microbiological analyses and to obtain representative samples for each treatment to be analysed, correct soil sampling in the laboratory, greenhouse trials, and field experiments is a critical step [22]. Temporal and spatial aspects could also be considered during sampling. To reduce the environmental impact, the cultivation of a specific crop over a consistent number of years and in the same soils is the most prominent factor [29]. The simplest approach used to overcome spatial variables is to collect the soil samples at random, ensuring that each sample has the same opportunity to be selected. For rhizosphere studies, soil attached to the roots should be carefully removed and collected. To evaluate external and internal root colonization, plant roots should be washed in sterile water or phosphate buffered saline and then homogenized in the same buffer. To study microbial endophytes, sterilization of the roots’ surface is necessary before carrying out the homogenization procedure. The procedure widely used to evaluate the persistence of inoculated microorganisms in soil and/or rhizosphere is the culture-dependent method [22]. However, with this method, it is not possible to evaluate the totality of microorganisms present in the soil and/or rhizosphere, because only 0.1 to 1.0% of soil microorganisms are culturable [30]. Moreover, this method is useful if the experiments are carried out in sterile conditions, without the interference of soil autochthonous microbial populations. Culture-based methods may be complemented with culture-independent approaches, such as PCR-based methods and next-generation sequencing, to examine the variations in the microbial community after inoculation treatment [22,31]. Therefore, a combination of culture-dependent methods and molecular approaches could be used to track inoculated strains or microbial consortia in natural habitats. The limiting step in both direct and indirect methods is the DNA extraction: due to the adhesion of DNA to soil particles, the extracted DNA will mostly represent the dominant species. It has been observed that to fully exploit the abilities of PGPB, it is very important to reach and maintain a high-density population on the roots of the host plants [32].

Nevertheless, it is also possible that microorganisms never colonize the plant despite their high density. Furthermore, there are also cases in which microorganisms induce the plant to synthesize molecules necessary for their nutritional needs. A particular example concerns the really useful pathogen *Agrobacterium*. Through a rare interkingdom DNA transfer, this bacterium moves some of its genes into its host’s genome, thereby inducing the host cells to proliferate. The result is uncontrolled cell growth leading to a tumour or excessive production of roots. The proliferating plant tissues produce opines, which are compounds that *Agrobacterium* and a few other organisms can use as a source of nutrients [33,34]. The most commonly used method to detect bacteria inside plant tissues involves the labelling of the cells with fluorescent systems, such as GFP, and their detection by fluorescence microscopy [22]. However, the use of GFP-tagged microbial strains is not applicable in field trials since the tagged microbial cells could be released into the environment. In addition, the autofluorescence of the plant cell walls makes the visualization of labelled microorganisms difficult in situ. Therefore, efforts should be made to find a suitable system to track PGPB interacting with plants under field conditions.

## 4. Plant Phenotyping to Analyse the Effects of Beneficial Microbes

PGPB efficiency is also greatly affected by plant age and developmental stage [33].

These parameters have greater effects on microorganisms present in the plant compartments than in the soil [35]. However, the mechanisms that regulate the distribution and differentiation of microbial communities during the various phases of plant development in the field are not yet well known. The microbial differentiation may be influenced by plant factors (such as root growth, physiology, architecture, morphology, and exudates), environmental factors (such as air, dust, rainfall, and temperature), edaphic factors, and fertilization regimes [36]. Plant phenotyping technologies based on non-destructive image analyses are useful instruments for addressing and understanding the complex plant-environment dynamics represented. The information resulting from the application of these technologies can be exploited for the genetic improvement of crops or for the analyses of the beneficial effects of microbial communities’ presence [37,38,39,40,41,42,43]. The development of new innovative approaches (visible imaging, fluorescence imaging, thermal infrared imaging, imaging spectroscopy, and other techniques) allowed to reach accuracy and precision in phenotyping analyses [35,37,39,43]. In particular, visible imaging provides information regarding plant growth morphology and allows the analysis of changes in phenotypic characteristics and the plant’s biomass. Another technology commonly used for plant phenotyping is spectroscopy imaging [35,38,43]. One of the parameters measured with this technology is the vegetation indices, the most popular of which is the normalized difference vegetation index, which is used to assess the general health status of crops. Early stress symptoms of plant diseases or the plant water status can be monitored through the use of thermal imaging, which is the most commonly used system for the analysis of stomatal activity, as the stomatal conductance increases with rising temperature [38,43]. Fluorescence imaging, consisting of the imaging of fluorescence signals obtained by illuminating samples with visible or UV (ultraviolet) light, is primarily used to study the effect of environmental conditions on photosynthesis and its associated metabolism [43]. Finally, modern optical 3D structural tomography and functional imaging techniques, such as Nuclear Magnetic Resonance Imaging and Positron Emission Tomography, have greatly improved living plant visualization in a non-destructive manner. Phenotyping can take place under laboratory, greenhouse or field conditions. In laboratory experiments, environmental factors can be controlled and varied during the experiments, but only a limited number of environmental factors can be investigated [42]. Anyway, laboratory experiments contribute to understanding specific plant dynamics in detail. This information should be then related to field conditions, which are more complex, highly variable and fluctuating in time and space, and therefore could better support scientists and breeders in the analyses of crop features [42,44,45,46].

## 5. Model Plants to Identify Plant and Microbial Candidate Genes Governing Plant-Microbe Interaction

The genetic features needed for an efficient association between plants and microbial communities are complex and still poorly understood. Even though it was observed that genotypic differences of the host plants significantly affect root associated microbial communities, plant breeding programs so far did not pay attention to the analysis of the microbial communities associated with them. In particular, no genetic loci affecting the establishment of association with the microbial communities have been identified. Basic studies have detected few plant molecular pathways so far, only those associated with shaping the plant-microbe interaction, and only for a few model cultivars [47]. It was shown that the plant-associated microorganisms are able to affect different plant traits, such as nutrient uptake, flowering time, and stress resistance [48]. Indeed, recent studies with microorganisms isolated from soil suggest that the capacity of plants to mobilize phosphorus (P) improves when they are associated with specific microbial communities [49]. Moreover, it was discovered that rhizosphere microbial communities influenced the flowering time of *Arabidopsis thaliana*, suggesting that microorganisms play a key role in plant functioning [50]. In addition, several studies concerning the beneficial effects of microorganisms on plant response to stresses, and in particular to the abiotic ones have also been reported [51,52,53]. Therefore, to modify specific plant traits, a combined approach of breeding plants and engineering the microbial community associated with them could be an effective approach [54,55]. However, due to the lack of integration between physiological data and those concerning the interaction with microbial communities, the targeted microbiome modification remains an arduous process. To design synthetic microbial communities for higher crop productivity, it is important to understand how plants and microbes communicate with each other, and which plant genes allow crops to shape the rhizosphere microbial community [56,57]. Modern technologies such as next-generation sequencing (NGS), omics approaches (metagenomics, transcriptomics, proteomics, metabolomics), and computational tools enable the understanding of molecular aspects of the plant-microbes interactions governing the plant traits. Recently, several reports investigated the influence of host genotype on different facets of the microbial communities. Genetic information about these interactions is becoming available for several crops and associated microbes [58,59]. In this regard, the CRISPR (clustered regularly interspaced short palindromic repeats)-based genome editing, is an ideal technology to obtain mutant plants or microbes differing from the parental ones only for a point mutation, without introducing exogenous DNA sequences [60,61,62]. The use of complementary sequencing and transcriptomics techniques currently available can lead to the production of a lot of sequence data, several mutants, even for the main crops, and to identification of specific genetic loci involved in plant-microbe interactions under field conditions. The CRISPR technology could exploit this information to introduce targeted genetic modifications in both plants and microbes, thus leading to the development of improved plants/microbes usable for sustainable agricultural practices [63,64,65]. However, a major obstacle to the genetic improvement of plants is represented by the lack of adequate legislation.

## 6. Application of Stress Conditions to Analyse Microbial Traits and Plant Phenotype under Conditions as Close as Possible to Those Found in the Open Field

The ultimate action of PGPB on plant growth and health depends on several factors, including the ability to survive, colonize, and establish interactions with the host plants.

The evaluation of these parameters is especially important for PGPB application in field conditions, where several variables hindering their success come into play.

The most commonly used approach for the selection of PGPB is to isolate the different strains and assess their growth promoting traits under aseptic conditions. However, this approach is limited since it does not take into account that in real conditions the PGPB efficiency depends not only on the individual traits analysed but also on their interaction with other factors, which can be associated with both plants and the environment. Therefore, to identify efficient PGPB, it is essential to evaluate as many variables as possible in greenhouse experiments, in order to simulate the real field conditions [66,67,68,69]. Moreover, considering that the environmental conditions have strong influence on the microbial communities associated with plants, having information about the place of origin of the tested strains can be helpful for field application [67,68]. The use of throughput sequencing of nucleic acids for molecular characterization of PGPB under real field conditions might be an effective approach to select PGPB candidates [69]. Plants living in extreme environments, such as areas characterized by high temperatures, high salinity, and low level of nutrients, have adapted to those conditions. This adaptation is the result of their genetic plasticity but is also due to the action of the microorganisms associated with them. Since these microorganisms have experienced the same adaptation as their host plants, they could represent an optimal source for the selection of PGPB usable to improve plant growth under real field conditions where they are frequently subjected to different abiotic stress conditions [67,70].

## 7. Effects of Climatic Conditions on Microbial Inoculants

It is now well recognized that climate changes negatively affect most of the living organisms of our planet, including plants and microorganisms [71]. The ever more sudden alterations in environmental conditions can induce changes in the physiology of plants, leading to a different distribution of fundamental elements, such as C and N, in the different compartments of the plants [71]. A different distribution of C and N within plants can cause alterations in the exudates released by the roots into the environment. The ability of beneficial bacteria, such as PGPB, to colonize the roots of the host plants is strongly influenced by root exudates, which have a complex composition and include molecules acting as chemoattractants. The recognition of root exudates represents the first step of the recruitment and colonization processes that allow PGPB to colonize the host plants. Therefore, climate changes negatively impact the composition of microbial communities in the soil and the activities of microbes interacting with the plants [72,73,74,75]. Moreover, climate changes can directly influence the microbial composition of soils. In particular, the scarcity of rain can lead to a reduction in the biomass of microorganisms present in the rhizosphere. Among them, there are PGPB, which help the host plants to counteract the negative effects caused by stress conditions (abiotic and biotic). As plant-associated bacteria depend on root exudates or plant metabolites and are substantially influenced by environmental parameters, it is plausible that these microbial communities will be increasingly affected by the extreme conditions associated with climate changes [76,77]. Therefore, understanding how climate influences, either directly by altering environmental conditions or indirectly by changing plant physiology, microbial community composition is a major challenge in the future, when the use of sustainable agriculture will be imperative.

## 8. Find the Best Practice for Application of PGPB in the Field

An important parameter that must be evaluated before using bioinoculants in agricultural practices is its vitality during the storage and application steps.

To ensure the activity of bio-inoculants, the use of microbial consortia, instead of single strains, to inoculate the host plants can be useful. Applying consortia, there will be a good probability that at least some of the microorganisms contained in the bioinoculants will survive [78]. However, even in the case of microbial consortia, it is important to evaluate how the environmental stresses can affect the vitality of bioinoculants. To this aim, various parameters, including bioinoculants formulation and delivery, should be monitored [79,80,81]. The identification of the right procedure for inoculant formulation is crucial to provide stabilization and protection of consortia during transport, storage, and application. This procedure should ensure the survival of a sufficient number of microbial cells able to exert their positive effect in the host plants. Bioinoculants industry has developed several systems that involve the use of different carriers (solid, slurry and liquid) and additives (i.e., adhesives and surfactants) [82]. Another important factor to consider is the release procedure, which can consist of the release of microbial suspensions (made in water, oils, or emulsions) directly into soils or applied on the seeds. Although the direct application in the soils allows to obtain a higher concentration of PGPB compared to the treatment on seeds [83], it is associated with a higher risk of contamination and to the loss of metabolic activities of PGPB contained therein [69]. Microbial encapsulation with polymeric hydrogels can be used to overcome these limitations. An alternative system is represented by solid formulations, in which solid carriers, such as peat, charcoal, vermiculite, cellulose and polymers, are used.

The use of alginate microbeads to encapsulate PGPB has yielded very promising results [84,85], as it is a reproducible method. However, even if this method is effective in laboratory experiments, it is economically disadvantageous when used on a large scale, such as open field applications. An alternative system, to increase inoculant survival rate and reduce contamination, could be the reduction of the moisture content of PGPB formulation by using a fluidized bed dryer (FBD) equipment. This system has a high drying rate, and the material is dried in a very short time and remains free-flowing and uniform [86,87]. One of the main disadvantages of this process is that it operates at about 37 °C to 40 °C, and therefore is more suitable for mesophilic organisms. To ensure a fair cell density during storage and to increase formulation efficiency, the alginate or FBD methods could be promising systems. To date, one of the main obstacles to the use of microbial-based bioinoculants is the lack of data regarding their efficacy across a range of agricultural field settings. Moreover, systems to evaluate the effective colonization of the host plants and the persistence of PGPB in soils over time have not yet been developed.

## 9. Functional Collaboration Combining Expertise in Basic Science, Development, Testing, and Marketing

To fully exploit the potentiality of PGPB as bio-inoculants for sustainable agriculture productions, the integration of biology, chemistry, physics, engineering, microbiology, biotechnology, genomics, computer science, and many other disciplines is a promising strategy. Using this approach, the selection of most suitable and effective strains can be performed for the production of high-quality bio-inoculants having higher performance and wide applicability. For example, in an area where phosphate utilization is the major concern, it is relevant to select the inoculant having the property of phosphate-solubilizing activity [88]. Moreover, the solid substrate-based bio-inoculants usually have a short shelf life (about 6 months) and are highly sensitive to direct sunlight, thus requiring specific storage conditions [10]. Therefore, the selection of bio-inoculants that maintain their potential efficacy beyond the period of 6 months and do not require temperature-controlled storage is desirable. Furthermore, since the lack of awareness regarding the use and role of bio-inoculants has hindered their mass spreading among the different stakeholders, the use of appropriate market information could be a successful strategy to overcome this gap [89]. To date, the demand for bio-inoculants is still not very high, and this is probably due to the implementation of policies that are either too restrictive or improper, especially in the countries of the European Community [90,91]. In the USA, APHIS regulates living organisms that could impact animal and plant health and that can be biocontrol organisms or have biopesticidal properties (https://www.aphis.usda.gov/aphis/home (accessed on 14 April 2022)). In Latin America, public policies are encouraging the use of alternatives to agrochemicals (pesticides and fertilisers) through the development of regulatory frameworks for product evaluation and approval. Colombia is the only country with specific legislation on biological inputs, while Argentina and Brazil have both recently developed a national programme to promote bio-inputs [92]. Therefore, the implementation of proper and uniform policies could favour the production of bio-inoculants and help to overcome the difficulties in registering new products.

## 10. Conclusions

To enhance plant protection and agriculture production in a sustainable way, there is a need for new tools to increase our knowledge on the mechanisms that regulate plant-microbiome interactions. In this context, an in-depth analysis, also based on model plants and microorganisms, will allow to better understand how microbes can contribute to the well-being of plants and how these beneficial effects can be harnessed for agricultural application. Furthermore, the development of specific and easy methodologies for the evaluation of PGP activities that inoculated strains have on the soil could help to clarify the processes that take place during plant-soil-microbe interactions. The potential application of beneficial microbes in agriculture seems unlimited, but more attention to the transfer of discoveries from lab to field, as well as technical, regulatory, and marketing issues, is mandatory. An integrated approach to address the multi-faceted nature of these challenges is most likely to succeed (Figure 1).

## Figures and Tables

**Figure 1 microorganisms-10-00865-f001:**
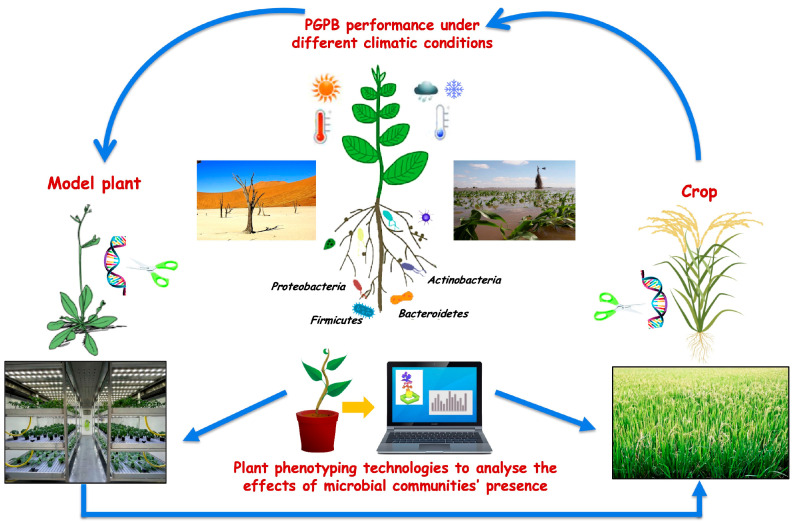
Plant Growth Promoting Bacteria (PGPB) for sustainable agriculture: link between laboratory and field experiments.

## Data Availability

Not applicable.

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
