# Peer review of "Exploitation of Plant Growth Promoting Bacteria for Sustainable Agriculture: Hierarchical Approach to Link Laboratory and Field Experiments"

_microorganisms, 2022, doi:10.3390/microorganisms10050865_

Round 1

Reviewer 1 Report

This Perspective manuscript presented by Massa et al covers most of the current concerns regarding the knowledge and applicability of PGP bacteria to improve sustainability in the agriculture. It is timely and discuss very well and in a clear manner some of the implications of these bacteria in agricultural systems.

Although this manuscript is fine, one should expect more emphasis in the importance of a proper taxonomic classification of those PGPB and also, the importance of the legislation, not only in Europe but worldwide.  

Thus, I would recommend to strengthen the discussions on these matters.

Two minor comments:

Line 97. ...they come...

Lines 132-134 please add ref for evidence.

Author Response

Reviewer 1

This Perspective manuscript presented by Massa et al covers most of the current concerns regarding the knowledge and applicability of PGP bacteria to improve sustainability in the agriculture. It is timely and discuss very well and in a clear manner some of the implications of these bacteria in agricultural systems.

Although this manuscript is fine, one should expect more emphasis in the importance of a proper taxonomic classification of those PGPB and also, the importance of the legislation, not only in Europe but worldwide.  

Thus, I would recommend to strengthen the discussions on these matters.

Authors’ reply - We acknowledge the Reviewer for her/his suggestions and introduced specific comments on taxonomic classification and legislation in the revised version of the manuscript (Lines at the end of section 2 and 9, respectively).

Two minor comments:

Line 97. ...they come...

Authors’ reply - We have considered your suggestion and modified the text in the revised manuscript accordingly. We also did a careful check of the entire manuscript and corrected other writing errors.

Lines 132-134 please add ref for evid

Authors’ reply - The revised manuscript contains an appropriate Reference for this topic.

Reviewer 2 Report

This manuscript is an overview of the different aspects governing the interaction between Plants and Plant Growth Promoting bacteria (PGPB). Challenges preventing the full exploitation of PGPB for sustainable agriculture are exposed. Perspectives to improve the use of PGPB for better yields are brought with a critical view of laboratory and field practices, as well as technologies for research and PGPB applications. Moreover, numerous recent publications are cited. I found the manuscript well structured, very informative and pleasant to read.

To provide a global visual information of the hierarchical approach to link laboratory and field experiments (title), I strongly suggest to add a schematic. In addition to provide a visual overview of the manuscript content, the schematic will likely increase the readership, as the reader will immediately see the different aspects of the proposed perspectives.

Minor comments:

  • Lines 27-32: references should be added to support these statements.
  • Typo mistakes are spread throughout the manuscript and should be fixed. Examples:

Line 9 : I think « million » should be replaced by “billion”.

Line 45: “mecchanisms” should be replaced by “mechanisms”

Line 100: “envoronmental » has to be replaced by “environmental”

Author Response

Reviewer 2

This manuscript is an overview of the different aspects governing the interaction between Plants and Plant Growth Promoting bacteria (PGPB). Challenges preventing the full exploitation of PGPB for sustainable agriculture are exposed. Perspectives to improve the use of PGPB for better yields file:///Users/carmenbianco/Downloads/microorganisms-1673413-peer-review.pdf are brought with a critical view of laboratory and field practices, as well as technologies for research and PGPB applications. Moreover, numerous recent publications are cited. I found the manuscript well structured, very informative and pleasant to read.

To provide a global visual information of the hierarchical approach to link laboratory and field experiments (title), I strongly suggest to add a schematic. In addition to provide a visual overview of the manuscript content, the schematic will likely increase the readership, as the reader will immediately see the different aspects of the proposed perspectives.

Authors’ reply - We agree with your comment, and we thank for the suggestion. A scheme of the proposed approach was introduced in the revised version of the manuscript as Figure (Figure 1).

Minor comments:

Lines 27-32: references should be added to support these statements.

Authors’ reply - The revised manuscript contains an appropriate Reference for this topic.

Typo mistakes are spread throughout the manuscript and should be fixed. Examples:

Line 9: I think « million » should be replaced by “billion”.

Line 45: “mecchanisms” should be replaced by “mechanisms”

Line 100: “envoronmental » has to be replaced by “environmental”

Authors’ reply - We have considered your suggestion and modified the text in the revised manuscript accordingly. We also did a careful check of the entire manuscript and corrected other writing errors.